# Development and validation of the Self-Awareness of Ego-Threatening Biases Questionnaire (SAETBQ)

Hanna Brycz[1], Aneta Chybicka[2], Paweł Smoliński[1], Mateusz Lammek[1], Andrzej Piotrowski[1], Sofia Hohmann[3], Nicola Baumann[3]*

1 University of Gdańsk, Gdańsk, Poland, 2 Pomeranian University in Słupsk, Słupsk, Poland, 3 University of Trier, Trier, Germany

* nicola.baumann@uni-trier.de

## Abstract

Awareness of social biases is crucial as they impact both individual behavior and societal outcomes. Whereas previous research indicates that self-awareness of ego-nonthreatening biases enhances self-regulation, the effects of self-awareness of ego-threatening biases remain underexplored. Preliminary findings suggest that awareness of ego-threatening biases related to rumination may lead to maladaptive states. However, these findings await replication with standardized instruments. To address this gap, we conducted two studies. In Study 1 ($N = 1609$), we developed and validated the 12-item Self-Awareness of Ego-Threatening Biases Questionnaire (SAETBQ). Consistent with our hypotheses, self-awareness of ego-threatening biases (as measured by the SAETBQ) correlated with higher moral disengagement, lower self-diagnostic motive, and lower integrative self-knowledge, indicating a tendency towards ego deterioration, whereas self-awareness of ego-nonthreatening biases (as measured by the Metacognitive Self questionnaire) showed the opposite pattern of correlations, indicating a tendency towards beneficial self-regulation. In Study 2 ($N = 681$), Dark Triad traits correlated positively and Light Triad traits negatively with self-awareness of ego-threatening biases. These results underscore the complex role of self-awareness in managing cognitive biases.

## Introduction

The main aim of the present research is to establish a new tool that accurately and reliably assesses human self-awareness of ego-threatening biases. The second goal directs our effort to understand the distinction between self-awareness of ego-threatening biases and self- awareness of ego-non threatening biases. Self-awareness of biases assumes metacognitive processes, meaning knowledge about one's own mental states, beliefs, and emotions. Self-awareness of ego-nonthreatening biases is defined as being aware of how self-regulatory biases and psychological

**Data availability statement:** The data of the present studies together with supplementary materials are available in the Open Science Framework repository at https://osf.io/vhwb5/?view_only=185af0d49b2946c2b-370b2289221494e.

**Funding:** The publication was supported by the Open Access Fund of Universität Trier and by the German Research Foundation (DFG).

**Competing interests:** The authors have declared that no competing interests exist.

rules (like classic conditioning) influence one's own behavior. Self-awareness of ego-threatening biases (like dehumanization) analogously is defined as being aware of how the biases that negatively assert about the perpetrator (threatening the ego) influence one's own behavior. Since previous research showed self-regulatory functions of high (vs. low) self-awareness of ego-nonthreatening biases (metacognitive self), we suspected the reverse pattern for self-awareness of ego-threatening biases. For the latter, a high level of self-awareness of ego-threatening biases may relate to potential deeper and accurate self-knowledge about the dark aspect of the self.

**Literature review**

**The nature of biases.** Biases arise from cognitive shortcuts, or heuristics, that are deeply ingrained in our decision-making processes and often lead to systematic errors and irrational behaviors, significantly impacting our interactions and perceptions of others [1–3]. The study of biases has long intrigued social psychologists, with pioneering work by Daniel Kahneman and Amos Tversky in the early 1970s laying the foundation for understanding the pervasive impact of heuristics on human cognition. Tversky and Kahneman [4] and Chen and colleagues [5] revealed that biases result from the fast, automatic, and intuitive thinking. Therefore, individuals are not necessarily aware of their biases.

Individuals who focus more on positive than negative information about options, for example, are biased (e.g., optimistic bias; [6]). Since biases are cognitive failures [7,8] they influence individual behavior but also have broader implications for societal outcomes. For example, biases can affect hiring decisions, interpersonal relationships, and even policy-making [9]. When individuals become aware of their own biases, this self-awareness can affect their self-regulation [10], self-concept [11,12], and social interactions [13]. However, different types of biases may affect individuals differently.

Thus, a separate problem is the issue of bias types itself. Research is frequently limited to one or several biases. To our best knowledge, the typology of biases is understudied [14]. Several studies focus on positive versus negative interpretation biases. Baumgardner and colleagues [15] created a measure for positive and negative interpretation biases. Hannuschke and colleagues [16] define personality-congruent perception bias as positive when it reflects a perception bias: neuroticism predicted participants' ratings of their interaction partners' sociability and warmth indicating a positive but not a negative bias in interpersonal perception. Positive perception bias may result from a personality-congruent contrast effect: neurotic people see themselves as less sociable than others. Authors did not find a personality-congruent selection effect (negative biases). Insufficient psychological literature on negative versus positive roles of biases in many contexts of human behavior make us enlarge the potential theoretical basis.

Mentioned above there are a few approaches that formulate distinct types of biases [15,16]. These approaches primarily classified biases based on the mechanisms that give rise to them and their societal consequences. In our research, we propose a new typology by classifying biases based on their effects on individuals

once they become aware of their biases. We identified two types of biases: ego-nonthreatening biases, conceptualized as the Metacognitive Self, and ego-threatening biases. We focus on whether awareness of the bias is positive or negative for the individual and how this awareness influences self-regulation. Positive biases can enhance social interactions and well-being as, for example, in positivity bias. Conversely, negative biases can diminish social interactions and lead to behaviors such as self-handicapping or behavioral consequences of dehumanization. Awareness of negative biases can result in self-threats and lower self-regulation, whereas awareness of positive biases fosters positive emotions, well-being, and self-regulation. We term this awareness of positive biases as the metacognitive self, based on previous literature, while our exploration of the consequences of awareness of negative biases is novel.

Thus, in our research, we propose a new typology by classifying biases based on their effects on individuals once they become aware of their biases. We identified two types of biases: ego-non threatening biases, conceptualized as the Metacognitive Self, and ego-threatening biases.

**Self-awareness of ego-nonthreatening biases (metacognitive self) and ego-threatening biases.** Self-awareness of one's emotions, thoughts, and decisions imply metacognition, which is the understanding and regulation of one's own cognitive processes. This concept dates back to Dewey [17] and has been expanded by scholars like Nelson and Narens [18]. Carden and colleagues [19] provided a comprehensive definition of self-awareness based on a systematic literature review: "Self-awareness consists of a range of components, which can be developed through focus, evaluation, and feedback, and provides individuals with an awareness of their internal state (emotions, cognitions, physiological responses), that drives their behaviors (beliefs, values, and motivations) and considers how these impact and influence others."

Building on the foundation of self-awareness, the concept of the Metacognitive Self (MCS) has been proposed. It reflects an individual's ability to understand and manage their cognitive processes and biases, promoting better self-knowledge and self-regulation. The Metacognitive Self is consistent with Taylor and Brown's [20] idea that some biases can be motivationally adaptive, while also acknowledging Block and Colvin's [21] limitations of this theory. It embraces the Aristotelian idea of the golden mean, balancing the adaptive role of psychological rules with accurate self-reflection.

Individuals with a high Metacognitive Self can better regulate their behavior by recognizing and managing their biases, leading to more consistent alignment with personal and social standards. High Metacognitive Self promotes deeper self-reflection and accurate assessment of one's actions, which is crucial for personal growth and development. The balance achieved through Metacognitive Self helps individuals maintain motivation without falling into the traps of overconfidence or excessive self-criticism. With high Metacognitive Self, individuals can internalize self-knowledge, making it readily accessible and applicable in various situations, thus enhancing decision-making and behavior [22,23].

Kleka and colleagues [24] expanded on the role of metacognitive self and intrinsic epistemic motivation in shaping an individual's awareness of biases. They suggest that feedback from emotional responses and interpersonal relationships helps individuals recognize biases within their own behavior. Similarly, Brycz and colleagues [25] found that strong self-awareness of positive biases (a high Metacognitive Self) fostered participants' willingness to learn social adaptive behavior even in the face of failure in contrast to participants with a low Metacognitive Self.

Although the psychological literature on the causes and nature of biases is rich (e.g., [19,26–28], research has mainly focused on self-awareness of biases that do not threaten individuals' ego (e.g., overgeneralizing positive impressions of others). Many findings show that higher self-awareness of ego-nonthreatening biases is associated with self-regulatory outcomes [29,30]. In contrast, individuals may also become aware of biases that threaten their self-esteem and moral self-views (e.g., dehumanizing the victim, prejudice). Preliminary findings indicate that self-awareness of ego-threatening biases is associated with negative outcomes for the individual [31] and teaching the nature of prejudice results in no outcome [32]. However, self-awareness of ego-threatening biases has rarely been investigated and systematically distinguished from self-awareness of ego-nonthreatening biases so far.

To close this gap, we developed and validated the self-awareness of ego-threatening biases questionnaire (SAETBQ). We predicted that self-awareness of ego-threatening biases is associated with defense mechanisms, a disjointed self-concept, and unfavorable social interactions, whereas self-awareness of ego-nonthreatening biases as measured by the Metacognitive Self (MSC) questionnaire is associated with self-regulation, an integrative self-concept, and favorable social interactions. Furthermore, we predicted Dark Triad traits (e.g., narcissism) to be positively and Light Triad traits (e.g., faith in humanity) to be negatively associated with self-awareness of ego-threatening biases.

**The importance of self-awareness of biases – metacognitive approach.**  Self-awareness implies metacognition, which is the understanding of one's own cognitive processes [17,18]. Thus, metacognition requires not the automatic but the slow, deliberate, and rational thinking [4,5]. Previous research has assumed that improving rational awareness of personal biases will mitigate the effects of automatic biases. But so far research revealed mixed results. Whereas Azad and colleagues [33] showed that enhanced self-awareness through feedback helps individuals recognize and understand the automatic processes that form the basis of implicit biases, Quillin and colleagues [34] found only small effects of prior cultural competency training on prejudice towards Black individuals.

To better understand how self-awareness of personal biases affects individuals, self-awareness theory offers valuable insights. The theory suggests that self-awareness initiates self-regulation [35]. Accordingly, when individuals recognize and become aware of the cognitive processes underlying their biases, they actively engage in self-regulation. Brycz and colleagues [25] demonstrate that self-awareness of personal ego-nonthreatening biases enhances self-improvement through self-regulation. Szczepanik and colleagues [36] showed smooth brain emotion regulation increasing with self-awareness of biases (i.e., metacognitive self). However, self-awareness theory does not specify the various strategies individuals may use. We assume that awareness of different types of biases initiate the use of distinct strategies.

**Effects of bias types on individuals.**  The typology of biases, as we mentioned above, is understudied. Single biases are in the eye of investigators. For example, the Halo Effect describes the cognitive error that when someone is perceived as warm, they appear more approachable, leading to more favorable interactions. Awareness of the Halo Effect bias allows individuals to engage positively with those who seem warm and to consciously adjust their reactions in a more positive manner to those who appear cold [37]. Another example is the fundamental attribution error, which attributes others' behaviors to personality traits rather than situational factors. Awareness of this bias encourages individuals to consider external factors influencing others' behaviors, fostering empathy and patience in social interactions [38].

Previous literature on self-awareness of biases (e.g., Metacognitive Self) has focused mostly on positive biases. These biases do not pose a threat to the ego because they maintain or enhance self-regulation. We propose that there are biases that pose a threat to the ego when individuals become aware of them and introduce them as ego-threatening biases.

**Self-awareness of ego-threatening biases.**  Previous literature shows that biases can have a negative impact on individuals and pose a threat to their ego [39]. These biases include negative rumination bias, which involves persistently thinking about negative feelings and emotions [40–43] and self-handicapping, where individuals create obstacles to their success to protect their self-esteem in the event of failure [44]. Another example is defensive projection [45,46] which involves attributing one's own unacceptable traits or choices to others, which can lead to interpersonal conflicts and hinder self-improvement. Thus, biases can pose a threat to individuals' ego.

Few studies have shown that biases, especially those that individuals have become aware of, have a negative impact on individuals. Pronin and Kugler [47] found that heightened awareness of one's own biases can lead to increased stress and discomfort, as individuals struggle to reconcile their biased behaviors with their self-perception as unbiased. Similarly, Wells and Matthews [42] discuss how increased self-awareness of negative biases can exacerbate emotional dysfunctions, such as anxiety and depression, by fostering maladaptive rumination and self-criticism. Therefore, empirical research indicates that individuals who are aware of personal biases that pose a threat to their ego may prefer strategies such as defense mechanisms over self-regulation.

The negative role of self-awareness of ego-threatening biases on human functioning is supported by theories of cognitive dissonance [48] and self-enhancement motives [49]. High self-awareness of ego-threatening biases leads to a cognitive dissonance between negative information about oneself in the form of ego-threatening biases and self-enhancement motives. This forces individuals to balance the negative insights about themselves with their need for positive self-esteem. One way to maintain positive self-esteem while being aware of ego-threatening biases is to use defense mechanisms (e.g., avoiding self-diagnostic information) that protect the ego but also promote a disjointed self-concept. For example, Forster and colleagues [50] showed that for people with a prevention focus, stereotype disconfirmation is a threat to efficient and effective self-regulation when the disconfirmation is discrepant from stereotypic beliefs they endorse. This produces negative emotions, vigilance motivation, and higher attention to both the disconfirming target and its background location.

Furthermore, individuals who use defense mechanisms are more likely to neglect moral standards which foster unfavorable social interactions characteristic of the dark triad ( [51,52]. The dark triad describes individuals that have an inflated sense of self-importance and a lack of empathy for others (Narcissism), use manipulative strategies and a focus on personal gain at the expense of others (Machiavellianism), and act impulsively and lack remorse or guilt (Psychopathy) [53,54]. Because the dark triad and self-awareness of ego-threatening biases both lead to more destructive and antisocial behaviors, they are likely to show positive correlations. The traits from the light triad, in contrast, emphasize faith in humanity, humanistic ideals, and Kantian moral standards and are likely to show negative correlations with self-awareness of ego-threatening biases.

## Problem and hypotheses

We have presented a theoretical background for the idea of a distinction between self-awareness of ego-threatening biases and self-awareness of ego-nonthreatening biases (metacognitive self). This distinction may help us to determine different paths and shed light on the beneficial versus harmful nature of bias awareness in creating self-construct, relations with other people, and our way of feeling and thinking. Researchers have attempted to mitigate the impact of negative biases, such as stereotyping and prejudice, on human behavior. For example, implicit bias training programs have been developed to educate individuals in influential positions about the harmful effects of implicit biases on society. Koenig-Robert and colleagues [55] demonstrated that such training can effectively reduce biases from subliminal sensory primes, with effects lasting up to one week. This highlights the importance of self-awareness of biases and targeted interventions in reducing their negative impact on both individual and societal levels. The rationale behind bias education is the belief that understanding one's own biases can prevent future errors. However, acknowledging ego-threatening biases can lead to defense mechanisms. This is particularly true for laypersons, as opposed to trained psychologists or those who have undergone implicit bias training. Thus, we predicted positive correlations between self-awareness of ego-threatening biases with moral disengagement, dark triad, less integrated self-concept in contrast to the negative (or none) correlations between enumerate variables and self-awareness of ego-nonthreatening biases (metacognitive self).

**Hypothesis 1: Self-Awareness of Ego-Threatening Biases (SAETB)**

Self-Awareness of ego-threatening biases is associated with higher moral disengagement (1a), lower self-diagnostic motives (1b), and less integrated self-concept (1c).

**Hypothesis 2: The Metacognitive Self (MCS)**

Self-awareness of ego-nonthreatening biases (i.e., the metacognitive self) is associated with lower moral disengagement (2a), higher self-diagnostic motives (2b), and more integrated self-concept (2c).

**Hypothesis 3: SAETB and the Dark Triad**

Self-Awareness of ego-threatening biases is positively associated with the traits of the Dark Triad: Narcissism (3a), Psychopathy (3b), and Machiavellianism (3c).

**Hypothesis 4: SAETB and the Light Triad**

Self-Awareness of ego-threatening biases is negatively associated with the traits of the Light Triad: Faith in Humanity (4a), Humanism (4b), and Kantianism (4c).

## The current studies

In two studies, we explored whether self-awareness of two types of biases affect individuals differently. While previous studies have highlighted the positive outcomes of self-awareness of nonthreatening biases (e.g., Metacognitive Self), the impact of self-awareness of ego-threatening biases remains underexplored. In Study 1, we investigated how the Metacognitive Self and self-awareness of ego-threatening biases influence individuals' self-regulation, self-concept, and social interactions. First, we developed a questionnaire to assess self-awareness of ego-threatening biases and investigated its effects on individuals. In Study 2, we examined how traits from the Dark and Light Triad relate to self-awareness of ego-threatening biases. Through our research, we aimed to contribute to the understanding of different types of biases and their impact on individuals, with a particular focus on biases that pose a threat to the ego.

## Study 1: Questionnaire development

The aim of Study 1 was to develop and validate a questionnaire for measuring self-awareness of ego-threatening biases (SAETB). SAETB, as defined in the introduction and outlined in the theory section, is the metacognitive awareness of how biases that reflect negatively on oneself influence one's behavior and thinking. Theoretically, it complements Metacognitive Self (MCS) – the awareness of biases that are ego-nonthreatening. In summary, SAETB involves acknowledging potentially uncomfortable truths about oneself (i.e., "I am biased, and these biases reflect badly on me"), whereas MCS involves acknowledging biases without this negative self-reflection. This definition provides important guidelines for operationalizing the SAETB psychometric model. Based on this definition, SAETB is conceptualized as a one-factor model with questionnaire items indicating awareness of various ego-threatening biases. The final score of the SAETBQ (Self-Awareness of Ego-Threatening Biases Questionnaire) indicates the final level of this awareness. A high SAETBQ score indicates more accurate self-knowledge about one's own negative biases, while a low score indicates limited recognition of these biases. Similarly, the related construct of Metacognitive Self is also operationalized as a single-factor construct where higher MCS scores indicate greater awareness and lower MCS scores indicate limited awareness [22].

With this operationalization in mind, we created an initial pool of items based on theoretical foundations and expert evaluations, then refined and validated these items through rigorous psychometric analysis to achieve a one-factor SAETBQ with the best and most reliable indicators. Below, we describe the process of selecting items and validating them against related constructs to demonstrate that SAETBQ shows the opposite pattern of correlations compared to self-awareness of ego-nonthreatening biases (see H1 and H2).

## Participants

Study 1 had 1,609 Polish adults aged 18–77 (M = 27.85, SD = 11.04) as participants. The sample had 950 females (*M* age = 28.15, *SD* = 11.30), 628 males (*M* age = 22.13, *SD* = 4.14), and 31 who did not report gender. The sample's education level matched the Polish population, with 20.4% having primary, 24.5% basic vocational, 35.6% high school, and 19.7% higher education. All participants answered questionnaires using Likert-type scales online via Google Forms (Google LLC., Mountain View, CA, USA) from November 19, 2021, to January 25, 2023. Participants were informed that the study was being conducted for scientific purposes and that their participation was both anonymous and voluntary. They were provided with details regarding the composition of the research team, along with contact information for the investigators. Participants were explicitly informed of their right to withdraw from the survey at any time. Informed consent was obtained through the first item of the questionnaire, which asked: *"Do you consent to participate in this study and acknowledge that you may withdraw at any point?"* Only adult participants took part in

the study. No sensitive data or any information that could identify the participants was collected. We used snowball sampling to recruit participants. We have obtained approval from the ethics committee of the University of Gdansk (certificate no. 71/2021/WNS).

## Measures

**Self-Awareness of Ego-Threatening Biases Questionnaire (SAETBQ).** The initial item pool consisted of 129 positive and negative biases. These items were carefully selected based on their relevance to the construct of bias awareness and were drawn from previous research conducted by Brycz [56]. The SAETBQ initial item pool reflects everyday scenarios and thoughts, such as the dehumanization of the victim, exemplified by items like: *When I treat someone badly, I think they deserve it because they are worse than me.* Participants rated each item on a 6-point Likert scale ranging from 1 (*strongly disagree*) to 6 (*strongly agree*).

**Metacognitive Self (MCS) questionnaire.** We used the 24-item self-report questionnaire to assess the metacognitive self (MCS-24; [22]). Each item describes a behavioral aspect related to biases that foster self-regulatory functions and provide positive assertions about the self. An example item for positivity bias is: *I tend to assess others and myself higher than we really are.* Participants rated each item on a 6-point Likert scale ranging from 1 (*strongly disagree*) to 6 (*strongly agree*). The scale showed good reliability in this study with Cronbach's $\alpha = .803$.

**Self-diagnostic motive scale.** We used the self-diagnostic motive scale [25] to measure the drive to seek diagnostic information about the self. It includes 6 items across 3 subscales assessing the effects of one's actions (e.g., *What negative things does the test say about me?*; Cronbach's $\alpha = .673$), seeking self-improvement (e.g., *What can I do to make myself better off in my life achievements?*; *Cronbach's* $\alpha = .624$), and comparing one's results with others (e.g., *To what extent did I complete the task worse than others?*; *Cronbach's* $\alpha = .763$). Responses are rated on a 6-point Likert scale from 1 (*I definitely don't want to know anything about it*) to 6 (*I definitely want to know about it*). In the present study, internal consistency of the total scale was Cronbach's $\alpha = .865$.

**Integrative self-knowledge scale.** We used the Polish adaptation [57] of the integrative self-knowledge scale [58] to measure an individual's tendency to unify self-experiences over time. It includes 12 items covering past experiences (e.g., *What I have learned about myself in the past has helped me to respond better to difficult situations*), present awareness (e.g., *Often, I am unaware of my thoughts and feelings as they are happening, and only later get some idea about what I may really have been experiencing*), and future goals (e.g., *By thinking deeply about myself, I can discover what I really want in life and how I might get it*). Each item is rated on a 5-point Likert scale from 0 (*largely untrue*) to 4 (*largely true*). In the present study, internal consistency was Cronbach's $\alpha = .745$.

**Moral disengagement.** We used the 8-item moral disengagement scale [59] to capture cognitive mechanisms that facilitate unethical behavior, such as moral justification (e.g., *It is okay to spread rumors to defend those you care about*) and displacement of responsibility (e.g., *People shouldn't be held accountable for doing questionable things when they were just doing what an authority figure told them to do*). Items are rated on a 7-point Likert scale from 1 (*strongly disagree*) to 7 (*strongly agree*). In the present study, internal consistency was Cronbach's $\alpha = .831$.

## Scale development

The development of the Self-Awareness of Ego-Threatening Biases Questionnaire (SAETBQ) proceeded through three stages. All details of the procedures, analyses, datasets, and additional examinations can be found in the Supplementary materials https://osf.io/vhwb5/?view_only=fa592ede448042bda9053cb1b565c5aa.

**Selection of biases.** The first stage of our investigation involved a selection of negative biases from the initial item pool based on the dimension of ego-threat: how much does being aware of a given bias threaten an individual's self-image? The initial item pool consisted of 129 well-known and popular biases (attribution biases, decision-making biases, and logical thinking biases like base rate neglect, and so forth) which 11 trained psychologists evaluated on mentioned

dimension using a scale from −5 (*"extremely ego-threatening"*) to +5 (*"not at all ego-threatening"*). This evaluation process had already been carried out in a previous study by Brycz [56], where the biases were scored on the dimensions but not selected for the SAETBQ. The evaluation judgments were highly consistent, as indicated by a strong Kendall's tau coefficient [56]. Out of the original 129 biases, we identified 55 that had a high potential to threaten one's ego. We suggest that a bias should have a score of at least −1 from the judges to be considered as ego-threatening. These biases were likely to damage one's self-image if admitted. We turned each of them into an item that reflected daily situations, thoughts, or feelings.

**Psychometric item selection.** In the second stage, we subjected a set of 55 biases identified in the previous stage to further analysis. We divided our sample of 1,609 participants into a training set (80% − 1,287 participants) and a test set (20% − 322 participants) [60,61]. The psychometric item selection was conducted exclusively on the training dataset using an exploratory factor analysis (EFA) with repeated data splits [62]. In each random data split, 60% of the training data was selected to assess the factor loadings of the 55 items on a single factor. Items with loadings meeting or exceeding 0.45 were retained. The random data splits were repeated in 1000 iterations. Items that met the 0.45 loading threshold in at least 80% of iterations were deemed most robust and retained. Next, we conducted confirmatory factor analysis (CFA) on the training dataset to validate the selected items on independent data and evaluate its fit using common model evaluation measures [63–65]. We also conducted additional analyses including bootstrap EFA and exploratory analysis of multifactor solutions. As these are additional exploratory analyses, their details and discussions are provided in the Supplementary materials.

**Construct validity.** The final stage involved conducting a correlation analysis to examine the relationships between the SAETBQ and various psychological constructs. We expected a positive correlation between self-awareness of ego-threatening biases, as measured by *SAETBQ,* and Moral Disengagement. This correlation would indicate that individuals who rationalize unethical behavior to protect their self-image are more likely to be aware of their negative biases (H1a). Conversely, we hypothesized negative correlations between self-awareness of ego-threatening biases and self-regulation indicators such as the Self-Diagnostic Motive Scale and its subscales (H1b). Self-awareness of ego-threatening biases may undermine self-regulation, as individuals struggle to maintain a positive self-view. Specifically, we expected that individuals with high self-awareness of ego-threatening biases would score lower on self-diagnostic motives, indicating a difficulty in integrating this awareness into a coherent self-concept (H1c). Additionally, we expected self-awareness of ego-nonthreatening biases, as measured by the Metacognitive Self questionnaire (MCS-24), to show negative correlations with moral disengagement (H2a) and positive correlations the self-diagnostic motive (H2b) and the integrative self-concept (H2c). These correlations would suggest that individuals who are aware of their positive biases are better able to self-regulate their behavior and maintain a coherent self-concept.

## Results

### Psychometric Item Selection

Results of EFA with repeated data splits indicated 12 items to be retained in the final model. Table 1 presents the descriptive statistics and intercorrelations for the full set (combined training and test sets) of final SAETBQ items. These 12 items were identified within the training set. Subsequently, we performed Confirmatory Factor Analysis (CFA) on the test dataset to validate these 12 items. The CFA model fit indices exceeded the commonly accepted thresholds for a good fit ($\chi2 = 112.26$, df = 54, p = .002, CFI (robust) =.950, TLI (robust) =.939, RMSEA (robust) =.050, SRMR (robust) =.045). The CFI and TLI values were notably high, surpassing the 0.90 threshold, while the RMSEA and SRMR values fell well below the 0.05 upper limit, indicative of an excellent fit. The standardized factor loadings for these items, as shown in Table 2, were all above .45. We also reran the CFA on the full dataset (combined training and test sets), with factor loadings presented in the last column of Table 2. The model fit measures for the full dataset also indicated an excellent fit ($\chi2 = 252.69$,

**Table 1. Means, standard deviations, and intercorrelations for the items of the Self-Awareness of Ego-Threatening Bias Questionnaire (SAETBQ) in Study 1 ($N=1609$; full dataset).**

| Items | *M* | *SD* | 1 | 2 | 3 | 4 | 5 | 6 | 7 | 8 | 9 | 10 | 11 |
|---|---|---|---|---|---|---|---|---|---|---|---|---|---|
| 1. SAETBQ26 | 3.13 | 1.41 | | | | | | | | | | | |
| 2. SAETBQ30 | 3.28 | 1.64 | .26 | | | | | | | | | | |
| 3. SAETBQ31 | 2.49 | 1.48 | .30 | .22 | | | | | | | | | |
| 4. SAETBQ32 | 2.58 | 1.34 | .35 | .33 | .40 | | | | | | | | |
| 5. SAETBQ33 | 2.85 | 1.49 | .36 | .37 | .38 | .43 | | | | | | | |
| 6. SAETBQ36 | 2.37 | 1.33 | .23 | .25 | .29 | .34 | .33 | | | | | | |
| 7. SAETBQ37 | 2.91 | 1.54 | .24 | .22 | .33 | .24 | .33 | .27 | | | | | |
| 8. SAETBQ38 | 3.02 | 1.53 | .27 | .26 | .36 | .33 | .32 | .29 | .32 | | | | |
| 9. SAETBQ42 | 3.13 | 1.45 | .30 | .20 | .30 | .28 | .31 | .24 | .25 | .30 | | | |
| 10. SAETB47 | 2.97 | 1.34 | .28 | .21 | .34 | .31 | .29 | .30 | .27 | .25 | .31 | | |
| 11. SAETBQ49 | 3.35 | 1.35 | .28 | .19 | .28 | .28 | .29 | .19 | .23 | .28 | .31 | .38 | |
| 12. SAETBQ55 | 2.98 | 1.47 | .29 | .26 | .29 | .31 | .40 | .33 | .31 | .27 | .24 | .28 | .31 |

**Table 2. Factor loadings for the items of the Self-Awareness of Ego-Threatening Bias Questionnaire (SAETBQ) in Study 1 ($N=1609$).**

| Bias Name | BN Item | Stand. Factor Loadings | |
|---|---|---|---|
| | | Test Set | Full Set |
| Defensive projection | **SAETBQ26.** I attribute my own unapproved characteristics to people whom I like and who are similar to me. | .570 | .530 |
| Self-handicapping | **SAETBQ30.** When I am not sure that I will succeed (e.g., on an exam), instead of working for it, I take actions (e.g., cleaning, playing) that will justify a possible failure. | .481 | .465 |
| Dehumanization of the victim | **SAETBQ31.** When I treat someone badly, I think they deserve it because they are worse than me. | .527 | .593 |
| Incorrect use of representative-ness heuristic | **SAETBQ32.** I explain extremely different phenomena by the same cause. | .618 | .614 |
| Bias in justifying decisions | **SAETBQ 33.** My decisions are often random and then I bend reality to justify my choice. | .671 | .650 |
| Discrepancy between behavior and values | **SAETBQ36.** I often find myself doing things that contradict my personal values. | .492 | .515 |
| Impact of lacking identity on aggression | **SAETBQ37.** Anonymity makes me more likely to violate moral or social norms. | .464 | .502 |
| Aggression generates aggression (in me) | **SAETBQ38.** When I behave aggressively towards a person, it increases the risk of being aggressive again towards that person and towards other people. | .514 | .542 |
| Egocentric thinking | **SAETBQ42.** I tend to see myself as playing a more central role in events than it actually is. | .516 | .506 |
| Overestimating one's participation in teamwork | **SAETBQ47.** When I work with others in a group, I overestimate my contribution to the work. | .570 | .534 |
| Availability heuristic | **SAETBQ49.** When a phenomenon concerns me, I overestimate its universality, and when it does not concern me, I underestimate its universality. | .613 | .498 |
| Social proof | **SAETBQ55.** I often do something just because other people do it. | .607 | .553 |

df = 54, p < .001, CFI (robust) =.955, TLI (robust) =.945, RMSEA (robust) =.046, SRMR (robust) =.032). The selected items demonstrated robust reliability across both samples. In the test dataset, internal consistency was Cronbach's α = .839 and McDonald's ω = .840 and these values remained consistent when analyzing the full dataset (Cronbach's α = .831; McDonald's ω = .832). These results indicate that the selected 12 items are the best indicators of our self-awareness of

ego-threatening biases construct. The final 12-item version of the SAETBQ is available in an English translation and in the Polish original (see Supplementary material).

## Construct validity

**Positive correlations.** As expected, the SAETBQ demonstrated significant positive correlations with constructs related to moral disengagement. As listed in Table 3, we found a significant positive correlation between SAETBQ and the Moral Disengagement scale ($r = .54$, $p < .01$), indicating that individuals who score higher on SAETBQ tend to engage in cognitive mechanisms that justify unethical behavior. This correlation supports our assumption that individuals who are more aware of their negative biases tend to rationalize unethical behavior (H1a).

**Negative correlations.** The analysis revealed negative correlations between SAETBQ and constructs associated with self-regulation and positive self-view (see Table 3). We found a significant negative correlation with the composite Self Motive scale ($r = -.08$, $p < .01$), suggesting that awareness of ego-threatening biases undermines the drive to seek diagnostic information about oneself. This finding is further supported by the significant negative correlations with the Self Evaluation subscale ($r = -.07$, $p < .01$) and the Self Improvement subscale ($r = -.16$, $p < .01$), indicating that awareness of ego-threatening biases was associated with a lower ability to assess one's actions and a reduced drive for self-improvement. However, the correlation with the Self Comparison subscale was not significant ($r = .01$, ns), indicating that self-awareness of ego-threatening biases does not impact self-comparison. The negative correlations align with those observed for the Integrative Self-Knowledge scale, which also showed a negative correlation with the SAETBQ ($r = -.56$, $p < .01$). Overall, these results suggest that individuals who are more aware of their negative biases struggle with self-regulation and maintaining a positive self-concept (H1b and H1c).

Although these correlations are small sized effects, they contrast with the correlations found for self-awareness of non-threatening biases as measured by the Metacognitive Self scale (MCS-24). Consistent with H2a, the Metacognitive Self scale correlated negatively with Moral Disengagement ($r = -.17$, $p < .01$). Consistent with H2b, the Metacognitive Self scale showed strong positive correlations with the composite Self Motive scale ($r = .46$, $p < .01$), as well as its subscales Self Evaluation ($r = .41$, $p < .01$), Self Improvement ($r = .45$, $p < .01$), and Self Comparison ($r = .33$, $p < .01$). Consistent with H2c,

**Table 3. Means, standard deviations, correlations, and 95% confidence intervals for positive and negative correlations in Study 1 ($N = 1609$).**

| Variable | M | SD | 1 | 2 | 3 | 4 | 5 | 6 | 7 |
|---|---|---|---|---|---|---|---|---|---|
| 1. SAETBQ | 2.92 | 0.86 | | | | | | | |
| 2. Metacognitive Self | 4.19 | 0.56 | −.01 | | | | | | |
| | | | [-.06,.04] | | | | | | |
| 3. Moral Disengagement | 2.68 | 1.13 | .54** | −.17** | | | | | |
| | | | [.51,.58] | [-.22, -.13] | | | | | |
| 4. Self Diagnostic Motive (SDM) | 4.54 | 1.09 | −.08** | .46** | −.18** | | | | |
| | | | [-.13, -.03] | [.42,.50] | [-.22, -.13] | | | | |
| 5. SDM: Self Evaluation | 4.58 | 1.31 | −.07** | .41** | −.18** | .89** | | | |
| | | | [-.12, -.02] | [.37,.45] | [-.22, -.13] | [.88,.90] | | | |
| 6. SDM: Self Improvement | 4.66 | 1.17 | −.16** | .45** | −.26** | .80** | .59** | | |
| | | | [-.21, -.11] | [.41,.49] | [-.30, -.21] | [.78,.82] | [.56,.62] | | |
| 7. SDM: Self Comparison | 4.36 | 1.36 | .01 | .33** | −.04 | .86** | .67** | .50** | |
| | | | [-.04,.06] | [.29,.37] | [-.09,.01] | [.85,.87] | [.64,.70] | [.46,.54] | |
| 8. Integrative Self-Knowledge | 2.29 | 0.67 | −.56** | .09** | −.39** | .16** | .14** | .23** | .04 |
| | | | [-.59, -.52] | [.04,.14] | [-.43, -.35] | [.11,.21] | [.09,.19] | [.19,.28] | [-.00,.09] |

*Note.* SAETBQ = Self-Awareness of Ego-Threatening Bias Questionnaire; * $p < .05$; ** $p < .01$.

the Metacognitive Self scale showed a significant positive correlation with the Integrative Self-Knowledge scale ($r = .09$, $p < .01$). These correlations support our assumption that the self-awareness of ego-nonthreatening biases has beneficial effects on the individual.

Interestingly, awareness of ego-nonthreatening and ego-threatening biases, as measured by MCS-24 and SAETBQ, respectively, appeared to be unrelated to each other ($r = -.01$, $p = .68$). This lack of correlation suggests that awareness of positive and negative biases operates independently within individuals. While awareness of positive biases contributes to self-regulation and a positive self-view, awareness of negative biases seems to undermine these aspects without influencing the other.

## Discussion

The findings of Study 1 show that we can reliably and validly assess self-awareness of ego-threatening biases with a 12-item questionnaire (SAETBQ). Our findings are consistent with the assumption that self-awareness of ego-threatening biases is associated with self-defense mechanisms and maladaptive outcomes for individuals as indicated by higher moral disengagement and a lower tendency to seek and integrate self-knowledge. Becoming aware of nonthreatening biases, in contrast, has the opposite effects for individuals and is associated with higher self-regulation. To further validate the SAETBQ and extend the correlational pattern to variables that may function as antecedents or driving forces rather than outcomes of bias awareness, we investigated the correlations with personality traits in Study 2.

## Study 2

The objective of this study was to test the personality correlates of the Self-Awareness of Ego-Threatening Bias Questionnaire (SAETBQ) as measured by the Dirty Dozen scale and the Light Triad scale. Specifically, we hypothesized that there would be positive correlations between the SAETBQ and the Dirty Dozen scale, with its subscales Narcissism (H3a), Psychopathy (H3b), and Machiavellianism (H3c). Conversely, we predicted negative correlations between the SAETBQ and the Light Triad scale, which includes Faith in Humanity (H4a), Humanism (H4b), and Kantianism (H4c).

### Participants

The study included $N = 681$ Polish adults aged 18–74 ($M = 33.97$, $SD = 12.681$). The mean age of the $n = 456$ female subsample was $M = 33.97$ years ($SD = 12.403$), and that of the $n = 218$ male subsample was $M = 34.68$ years ($SD = 13.215$). The educational structure of the sample was as follows: 0.6% had middle school education, 4.4% had vocational education, 44.3% had secondary education, 11.6% had a bachelor's/engineering degree, and 39.1% had a master's degree. The survey was conducted online, participants were informed that their participation was voluntary and that their responses would be anonymous and confidential. The survey was conducted between 18 February 2024 and 18 April 2024 using Google Forms (Google LLC., Mountain View, California, USA). Participants indicated their informed consent by clicking a consent button in the survey. Snowball sampling was used. We have obtained approval from the ethics committee of the University of Gdansk (certificate no. 71/2021/WNS).

### Measures

In addition to the 12-item Self-Awareness of Ego-Threatening Bias Questionnaire (SAETBQ) from Study 1 (Cronbach's α = .866; see supplementary material for English and Polish versions), we used the following questionnaires.

**Dirty dozen dark triad scale.** We used the dirty dozen scale [66] that was adapted for use in Poland [67]. It is a 12-item self-report questionnaire designed to assess Narcissism (e.g., *I tend to want others to admire me*; Cronbach's α = .832), Psychopathy (e.g., *I tend to lack remorse*; Cronbach's α = .628), and Machiavellianism (e.g., *I have used deceit or lied to get my way*; Cronbach's α = .842) with four items each. Participants were asked to rate the extent to which they

believe each statement accurately describes their own behavior, using a 5-point Likert scale from 1 (*not at all true*) to 5 (*extremely true*).

**Light triad scale.** We used the light triad scale by Kaufman and colleagues [68] that was adapted by Gerymski and Krok [69] for use in Poland to measure Faith in Humanity (e.g., *I tend to see the best in people*; Cronbach's $\alpha = .704$), Humanism (e.g., *I tend to admire others*; Cronbach's $\alpha = .617$), and Kantianism (e.g., *I prefer honesty over charm*; Cronbach's $\alpha = .598$) with four items each. Participants rated their level of agreement with each statement on a 5-point Likert scale from 1 (*strongly disagree*) to 5 (*strongly agree*).

## Results and discussion

We found support for all our predictions. The Self-Awareness of Ego-Threatening Bias Questionnaire (SAETBQ) shows positive correlation with all three aspects of the Dirty Dozen scale. As listed in Table 4, correlations range from $r = .32$ for Psychopathy over $r = .46$ for Narcissism up to $r = .52$ for Machiavellianism. This suggests that self-awareness of ego-threatening biases is used to exploit others and pursue one's own objectives, regardless of the other person's interests. The higher the score on the Dirty Dozen scale, the higher the self-awareness of ego-threatening biases. This might imply that self-awareness of such biases is not always used positively. Although self-awareness of biases in general is a desirable developmental trait, our study results indicate that it might also be driven by negative tendencies such as Machiavellian, Narcissism, and Psychopathy.

Conversely, correlations between the Self-Awareness of Ego-Threatening Bias Questionnaire (SAETBQ) and the Light Triad scale were negative. As listed in Table 5, correlations range from $r = -.08$ for humanism, over $r = -.15$ for faith in humanity, up to $r = -.39$ for Kantianism, which involves treating people as ends in themselves and avoiding using them as means. This seems to validate the accuracy of our scale, as Kantianism is theoretically the opposite of Machiavellianism, as reflected in our survey results.

## General discussion

Our study aimed at differentiating between self-awareness of positive (ego-nonthreatening) and negative (ego-threatening) biases and exploring the consequences linked to the latter. To do so, we developed a novel instrument: the Self-Awareness of Ego Threatening Biases Questionnaire (SAETBQ). It includes 12 items rated on a 6-point scale ranging from 1 (*strongly disagree*) to 6 (*strongly agree*). Each item describes a specific bias, such as defensive projection illustrated by the item: *I attribute my own unapproved characteristics to people whom I like and who are similar to me*. In Study

**Table 4. Means, standard deviations, correlations, and 95% confidence intervals between the Self-Awareness of Ego-Threatening Bias Questionnaire (SAETBQ) and the dirty dozen scale in Study 2 (*N* = 681).**

| Variable | *M* | *SD* | 1 | 2 | 3 | 4 |
|---|---|---|---|---|---|---|
| 1. SAETBQ | 2.53 | 0.82 | | | | |
| 2. Narcissism | 9.47 | 3.81 | .46** | | | |
| | | | [.40,.51] | | | |
| 3. Psychopathy | 7.65 | 2.87 | .32** | .29** | | |
| | | | [.26,.39] | [.22,.36] | | |
| 4. Machiavellianism | 7.21 | 3.24 | .52** | .56** | .53** | |
| | | | [.46,.57] | [.50,.61] | [.48,.58] | |
| 5. Dark Triad | 24.33 | 7.95 | .55** | .81** | .72** | .87** |
| | | | [.49,.60] | [.78,.84] | [.68,.75] | [.85,.88] |

* $p < .05$; ** $p < .01$

**Table 5. Means, standard deviations, correlations, and 95% confidence intervals between the Self-Awareness of Ego-Threatening Bias Questionnaire (SAETBQ) and the light triad scale in Study 2 (*N* = 681).**

| Variable | *M* | *SD* | 1 | 2 | 3 | 4 |
|---|---|---|---|---|---|---|
| 1. SAETBQ | 2.53 | 0.82 | | | | |
| 2. Faith in Humanity | 13.10 | 3.07 | −.15** | | | |
| | | | [-.22, -.07] | | | |
| 3. Humanism | 14.85 | 2.48 | −.08* | .41** | | |
| | | | [-.16, -.01] | [.35,.47] | | |
| 4. Kantianism | 15.45 | 2.55 | −.39** | .33** | .37** | |
| | | | [-.45, -.32] | [.26,.39] | [.30,.43] | |
| 5. Light Triad | 43.40 | 6.18 | −.27** | .80** | .76** | .72** |
| | | | [-.34, -.20] | [.77,.82] | [.72,.79] | [.69,.76] |

* *p* < .05; ** *p* < .01.

1, we established the reliability and validity of the Self-Awareness of Ego-Threatening Biases Questionnaire (SAETBQ) while also comparing it to the Metacognitive Self questionnaire, which measures self-awareness of ego-nonthreatening biases. In Study 2, we expanded our validity analysis by examining correlations between the SAETBQ and constructs from both the Dark Triad and Light Triad. In the following, we will discuss the findings along our two main constructs.

## Self-awareness of ego-threatening biases

We predicted that the Self-Awareness of Ego-Threatening Bias Questionnaire (SAETBQ) would show negative correlations with self-regulatory outcomes. Consistent with our assumptions, the SAETBQ significantly and negatively correlated with Integrative Self-Knowledge, indicating an inconsistent and problematic self. The SAETBQ also showed strong and negative relationships with the self-diagnostic motive scale and its subscales, which further supported the idea of a troubled self, worries about seeking self-knowledge, and low desire to improve oneself. Furthermore, we obtained a significant and positive correlation between the SAETBQ and moral disengagement. Moral disengagement is the psychological process of convincing oneself that moral standards do not apply in a given context, detaching ethical reactions from inhumane conduct to prevent self-condemnation [70].

The present findings suggest an additional defense mechanism driven by self-awareness of negative biases. Participants with higher self-awareness of ego-threatening biases accommodate the discrepancy between negative information about themselves (e.g., *when I treat someone badly, I think they deserve it because they are worse than me*) and the self-enhancement motive [71] by disengaging from moral standards. They are forced to reconcile negative information about their misconduct (having negative biases) and the desire to feel positive self-esteem. One way of resolving this conflict may be neglecting communal moral standards and focusing on agency [72], as communion and agency are negatively correlated in our minds and operating systems. When individuals become aware of their negative biases, the solution to maintain positive self-esteem may be focusing solely on one's agency while ignoring communal moral standards. This is a hypothesis for further investigation, especially in experimental rather than correlational studies.

The results of our Study 1 led us to predict positive correlations between the SAETBQ and the Dark Triad and a negative correlation with the Light Triad in Study 2. Our findings supported the predictions. The SAETBQ was highly and positively correlated with all three components of the Dark Triad, especially Machiavellianism. It seems that an immoral attitude towards others helps people to become aware of and use negative biases for getting their own way. This speculation finds further support from the negative and significant correlations between the Light Triad and the SAETBQ: More humanistic attitudes seem to prevent people from becoming aware of and/or utilizing negative biases.

### Self-awareness of ego-nonthreatening biases

We predicted that self-awareness of ego-nonthreatening biases as assessed by the Metacognitive Self (MCS) would be positively correlated with self-regulatory outcomes. We found a significant positive correlation between the Metacognitive Self and Integrative Self-Knowledge, replicating previous findings [22] and confirming that awareness of positive biases is beneficial for the individual. The positive correlation suggests that participants with higher levels of Metacognitive Self are motivated to improve their self-knowledge and achievements, aligning with earlier results [22,25]. Moreover, our results showed a negative correlation between Metacognitive Self and moral disengagement. This suggests that higher levels of Metacognitive Self enable individuals to regulate their moral behavior more effectively, as supported by Bandura's [73] social-cognitive theory of morality. According to this theory, self-regulatory mechanisms, including self-reflective and self-sanctioning processes, help individuals adhere to internalized moral standards. Our findings, in line with Kleka and colleagues [23], indicate that Metacognitive Self functions in tandem with self-efficacy and hope, highlighting its broad self-regulatory role.

### Theoretical and practical implications

The present research has important implications. The psychological literature states that self-awareness of implicit and explicit biases helps individuals be good, empathetic workers. Consistent with this view, some findings show successful implicit bias training [55]. However, other findings have revealed reversed effects of implicit bias training [74] or replication problems [75]. This may be due to the lack of differentiation between ego-nonthreatening and ego-threatening biases. Our findings show that they have virtually opposite effects. Thus, ego-threatening biases pose a challenge for implicit bias training and may require very safe and mindful context conditions.

Laypersons are likely to have not undergone implicit bias training, mindfulness training, or psychotherapy that might support integrative coping strategies for dealing with awareness of ego-threatening biases and their impact on self-esteem. In future research, it seems crucial to investigate how teaching self-awareness of ego-threatening biases in a safe and relaxing context will change the unwanted correlates of SAETBQ, such as moral disengagement, self-deterioration, and low motivation for self-improvement. In contrast, teaching Metacognitive Self and the awareness of ego-nonthreatening biases seems to always be a beneficial technique.

### Limitations and future directions

Our study's correlational design limits the ability to infer causality. Future research should include experimental and longitudinal studies to better understand the causal relationships between self-awareness of biases and various behavioral outcomes. We plan to conduct experiments and longitudinal studies to further investigate how increased self-awareness of positive biases as measured by the Metacognitive Self scale and negative biases as measured by the Self-Awareness of Ego-Threatening Biases questionnaire (SAETBQ) influences behavior over time. Another avenue for future research could be the inclusion of indirect measures to extend the behavioral effects of bias awareness. For example, greater awareness of negative biases might be associated with stronger defensive projection as measured by the tendency to blame others for one's own poor choices (e.g., misattributing unattractive self-chosen activities as external assignments; [45]). Finally, future research could explore the context conditions that increase or buffer the negative outcomes of self-awareness of ego-threatening biases. For example, negative outcomes may be stronger under experimental arousal of ego threat and negligible under relaxed conditions.

### Conclusion

The present research shows that people cannot only be aware of their positive but also of their negative biases. This awareness can be reliably and validly assessed with the 12 item Self-Awareness of Ego-Threatening Biases questionnaire

(SAETBQ) and functions relatively independent of awareness of positive biases. Higher awareness of negative biases is associated with lower integrative self-knowledge, higher moral disengagement, and higher traits of the Dark Triad, indicating the tendency to deteriorate the ego. Taken together, the present research opens an avenue for research on the complex role of self-awareness in managing cognitive biases and its implications for personal and social functioning.

## Supporting information

**S1 File. Self-Awareness of Ego-Threatening Biases Questionnaire (SAETBQ) in english and polish.** (PDF)

**S2 File. Supplementary material: OSF repository files overview.** OSF Link: https://osf.io/vhwb5/?view_only=185af0d49b2946c2b370b2289221494e. (PDF)

## Acknowledgments

We thank the anonymous reviewers for their valuable suggestions and discussions.

## Author contributions

**Conceptualization:** Hanna Brycz.

**Data curation:** Aneta Chybicka.

**Formal analysis:** Paweł Smoliński.

**Investigation:** Hanna Brycz, Aneta Chybicka, Mateusz Lammek, Andrzej Piotrowski.

**Methodology:** Paweł Smoliński.

**Project administration:** Aneta Chybicka, Mateusz Lammek, Andrzej Piotrowski.

**Software:** Paweł Smoliński.

**Supervision:** Hanna Brycz, Aneta Chybicka, Paweł Smoliński.

**Validation:** Aneta Chybicka.

**Visualization:** Hanna Brycz, Aneta Chybicka.

**Writing – original draft:** Hanna Brycz, Aneta Chybicka, Paweł Smoliński, Mateusz Lammek, Andrzej Piotrowski, Sofia Hohmann, Nicola Baumann.

**Writing – review & editing:** Hanna Brycz, Aneta Chybicka, Paweł Smoliński, Mateusz Lammek, Andrzej Piotrowski, Sofia Hohmann, Nicola Baumann.

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
