## [Decision Letter · Decision Letter 0]

PONE-D-25-01433Development and Validation of the Self-Awareness of Ego-Threatening Biases QuestionnairePLOS ONE

Dear Dr. Baumann,

Thank you for submitting your manuscript to PLOS ONE. After careful consideration, we feel that it has merit but does not fully meet PLOS ONE’s publication criteria as it currently stands. Therefore, we invite you to submit a revised version of the manuscript that addresses the points raised during the review process.

**For your revision, we kindly ask that you pay special attention to the following points, among other comments raised by the reviewers:**

**Separate the Introduction and Literature Review sections.****Add a subsection to clarify the hypotheses.****Reconsider the validity of the simplified one-factor model used in Study 1.****Temper the claims in Study 2, as they rely on the preliminary findings of Study 1.****Improve the reproducibility of the R code, including clearer documentation and environment specifications.**

We look forward to receiving your revised manuscript.

Kind regards,

Hidenori Komatsu

Academic Editor

PLOS ONE

**Journal requirements:** 1. When submitting your revision, we need you to address these additional requirements. Please ensure that your manuscript meets PLOS ONE's style requirements, including those for file naming. The PLOS ONE style templates can be found at https://journals.plos.org/plosone/s/file?id=wjVg/PLOSOne_formatting_sample_main_body.pdf and https://journals.plos.org/plosone/s/file?id=ba62/PLOSOne_formatting_sample_title_authors_affiliations.pdf 2. Please provide additional details regarding participant consent. In the ethics statement in the Methods and online submission information, please ensure that you have specified what type you obtained (for instance, written or verbal, and if verbal, how it was documented and witnessed). If your study included minors, state whether you obtained consent from parents or guardians. If the need for consent was waived by the ethics committee, please include this information. 3. Your ethics statement should only appear in the Methods section of your manuscript. If your ethics statement is written in any section besides the Methods, please move it to the Methods section and delete it from any other section. Please ensure that your ethics statement is included in your manuscript, as the ethics statement entered into the online submission form will not be published alongside your manuscript. 4. We note that this data set consists of interview transcripts. Can you please confirm that all participants gave consent for interview transcript to be published? If they DID provide consent for these transcripts to be published, please also confirm that the transcripts do not contain any potentially identifying information (or let us know if the participants consented to having their personal details published and made publicly available). We consider the following details to be identifying information:- Names, nicknames, and initials- Age more specific than round numbers- GPS coordinates, physical addresses, IP addresses, email addresses- Information in small sample sizes (e.g. 40 students from X class in X year at X university)- Specific dates (e.g. visit dates, interview dates)- ID numbers Or, if the participants DID NOT provide consent for these transcripts to be published:- Provide a de-identified version of the data or excerpts of interview responses- Provide information regarding how these transcripts can be accessed by researchers who meet the criteria for access to confidential data, including:a) the grounds for restrictionb) the name of the ethics committee, Institutional Review Board, or third-party organization that is imposing sharing restrictions on the datac) a non-author, institutional point of contact that is able to field data access queries, in the interest of maintaining long-term data accessibility.d) Any relevant data set names, URLs, DOIs, etc. that an independent researcher would need in order to request your minimal data set. For further information on sharing data that contains sensitive participant information, please see: https://journals.plos.org/plosone/s/data-availability#loc-human-research-participant-data-and-other-sensitive-data If there are ethical, legal, or third-party restrictions upon your dataset, you must provide all of the following details (https://journals.plos.org/plosone/s/data-availability#loc-acceptable-data-access-restrictions):a) A complete description of the datasetb) The nature of the restrictions upon the data (ethical, legal, or owned by a third party) and the reasoning behind themc) The full name of the body imposing the restrictions upon your dataset (ethics committee, institution, data access committee, etc)d) If the data are owned by a third party, confirmation of whether the authors received any special privileges in accessing the data that other researchers would not havee) Direct, non-author contact information (preferably email) for the body imposing the restrictions upon the data, to which data access requests can be sent 5. Please include captions for your Supporting Information files at the end of your manuscript, and update any in-text citations to match accordingly. Please see our Supporting Information guidelines for more information: http://journals.plos.org/plosone/s/supporting-information.

Reviewers' comments:

Reviewer's Responses to Questions

**Comments to the Author**

1. Is the manuscript technically sound, and do the data support the conclusions?

Reviewer #1: Yes

Reviewer #2: Partly

2. Has the statistical analysis been performed appropriately and rigorously? 

Reviewer #1: Yes

Reviewer #2: Yes

3. Have the authors made all data underlying the findings in their manuscript fully available?

Reviewer #1: Yes

Reviewer #2: Yes

4. Is the manuscript presented in an intelligible fashion and written in standard English?

Reviewer #1: Yes

Reviewer #2: Yes

5. Review Comments to the Author

**Reviewer #1:**  Dear Authors,

Congratulations on preparing an excellent manuscript. I thoroughly enjoyed reading your work and believe it offers meaningful and insightful contributions to the field. The newly developed questionnaire will undoubtedly open valuable avenues for future research. Both studies presented were thoughtfully designed, with appropriate data analysis methods. Additionally, your discussions related to the results are insightful and relatively comprehensive. However, I have several suggestions to further strengthen the literature review section.

First, the introduction and literature review sections currently lack clear separation and detailed elaboration. Although the research background was briefly introduced at the outset, gaps identified clearly, and key concepts such as biases and self-awareness introduced effectively, a clearer and more structured distinction between these sections would significantly improve readability and coherence.

Second, your literature review could benefit from deeper exploration, particularly regarding the connection between self-awareness and metacognition. Given the importance of the metacognition concept to your study, expanding the discussion on how self-awareness specifically relates to metacognition would greatly enhance the relevance and depth of your theoretical framing.

Furthermore, while you have successfully introduced the broader research context, highlighted existing challenges, and summarized past literature relevant to your hypotheses, the literature review would be even stronger if you included a critical analysis of past studies. Please consider explicitly discussing the strengths, limitations, and key lessons derived from previous research. This approach would better position your study within existing literature and underscore its contributions.

Lastly, I recommend dedicating a specific subsection to clearly outline your hypotheses. It would be particularly beneficial if you explicitly connect these hypotheses to the research gaps you aim to address, emphasizing how your research introduces novelty. Ideally, this subsection should serve as a concise summary highlighting the significance and rationale for conducting your research, clearly linking research gaps, hypotheses, and anticipated contributions.

Wishing you all the best with your publication.

Kind regards,

Reviewer

**Reviewer #2:**  Acknowledgments:

I would like to acknowledge that my expertise lies primarily in statistical modeling, psychometric validation, and methodological rigor rather than psychological theory or clinical interpretation. My review thus focuses explicitly on the statistical and methodological choices, rigor, transparency, and reproducibility of the analyses conducted. Interpretations of psychological constructs or theoretical frameworks should also be evaluated by reviewers whose expertise aligns more closely with these aspects.

Methodology: Overall Thoughts and Recommendations

The manuscript is partly technically sound: statistical analyses performed are correct given their methodological choices, but foundational methodological decisions (factor structure choice, transparency, reproducibility) require significant improvement.

1. The authors’ findings linking self-awareness of ego-threatening biases (SAETBQ) and moral disengagement are intriguing but potentially problematic due to methodological choices. Specifically, adopting a simplified one-factor solution despite clear statistical support for a four-factor structure risks oversimplifying complex psychological relationships, thereby influencing both the interpretation and practical implications of the results. Given the ethically sensitive nature of the conclusions drawn, the authors should explicitly acknowledge how their decision to use a single-factor model limits interpretive clarity, nuance, and generalizability. This limitation underscores the need for further investigation and caution before drawing definitive or potentially stigmatizing conclusions.

2. The authors’ claim that “the findings of Study 1 show that we can reliably and validly assess self-awareness of ego-threatening biases with a 12-item questionnaire (SAETBQ)” appears overstated, given significant unresolved methodological concerns. The claimed reliability and validity depend heavily on a simplified one-factor model selected despite robust statistical evidence favoring a more nuanced four-factor solution. Given that Study 2 relies entirely on this measure, the authors should explicitly temper this claim by clearly acknowledging the preliminary nature and methodological limitations of Study 1’s findings.

Computational Methods and Data Validation:

The authors have commendably made their dataset publicly available, significantly supporting transparency. However, the supplementary R code intended to enable independent reproduction of their analyses lacks critical details necessary for genuine reproducibility. Specifically:

1. Insufficient documentation and explanatory comments, making it difficult to follow the logic or interpret intermediate outputs clearly.

2. No specification of the R version, package dependencies, or computational environment (e.g., no use of standard reproducibility tools such as sessionInfo() or renv).

3. Absence of a clearly structured workflow or explicit instructions for execution, hindering straightforward reproduction attempts.

For the sake of transparency, rigor, and reproducibility—fundamental principles in statistical modeling and psychometric validation—these issues must be explicitly addressed. The authors are encouraged to substantially improve code documentation, explicitly provide computational environment details, and clearly structure the analytical workflow.

References:

A brief review of the manuscript’s references reveals a notable absence of citations to foundational methodological literature on Exploratory Factor Analysis (EFA), Confirmatory Factor Analysis (CFA), and the interpretation of standardized factor loadings. Including relevant references to standard psychometric and statistical modeling literature (e.g., Brown, 2015; Fabrigar & Wegener, 2011; Kline, 2015) would enhance methodological clarity and support the rigor and appropriateness of the analytical approaches used.

Recommended foundational sources the authors could cite:

Brown, T. A. (2015). Confirmatory factor analysis for applied research (2nd ed.). Guilford Publications.

Fabrigar, L. R., & Wegener, D. T. (2011). Exploratory factor analysis. Oxford University Press.

Kline, R. B. (2015). Principles and practice of structural equation modeling (4th ed.). Guilford Publications.

Incorporating these references would significantly enhance the manuscript’s credibility and alignment with standard psychometric practices.

Summary:

The foundational methodological limitations identified in Study 1—particularly the insufficient justification for adopting a one-factor solution despite clear statistical evidence favoring a four-factor structure, inadequate transparency regarding item selection and stability, and reproducibility concerns—my primary recommendation is that the authors revisit and rigorously re-evaluate Study 1 before drawing strong conclusions from subsequent analyses (e.g., Study 2). Addressing these foundational issues will significantly strengthen the psychometric validity, interpretability, and ethical clarity of the findings.

Additionally, while the authors’ commitment to open data practices is commendable, the supplementary code is inadequately documented and lacks necessary details for true reproducibility, significantly limiting independent verification of their results.

For these reasons I recommend major revisions to address the insufficient justification for choosing a one-factor model despite clear statistical evidence supporting a four-factor solution, inadequate documentation and reproducibility of analyses, and overstated interpretations based on preliminary findings. Addressing these issues thoroughly, including incorporating references related to foundational analytics, is critical to ensuring the manuscript’s methodological rigor, interpretability, and ethical clarity.

Enhanced documentation of computational methods is also necessary to ensure data reproducibility.

Consideration:

Although not currently mentioned in the manuscript, the authors might consider leveraging advanced analytical methods, such as machine-learning-driven exploratory techniques or automated cross-validation (e.g., AI-supported factor selection, stability assessments, or iterative modeling), in future validations of their scale. Such AI-enhanced approaches could potentially address methodological shortcomings by improving factor-selection rigor, transparency, reproducibility, and generalizability.

6. PLOS authors have the option to publish the peer review history of their article (what does this mean? ). If published, this will include your full peer review and any attached files.

**Do you want your identity to be public for this peer review?** For information about this choice, including consent withdrawal, please see our Privacy Policy .

Reviewer #1: No

Reviewer #2: No

---

## [Author Response · Author response to Decision Letter 1]

26 May 2025

Response to reviewers

Editor Comments: For your revision, we kindly ask that you pay special attention to the following points, among other comments raised by the reviewers:

1. Separate the Introduction and Literature Review sections.

2. Add a subsection to clarify the hypotheses.

3. Reconsider the validity of the simplified one-factor model used in Study 1.

4. Temper the claims in Study 2, as they rely on the preliminary findings of Study 1.

5. Improve the reproducibility of the R code, including clearer documentation and environment specifications.

Response: Thank you very much for your effort undertaken to revise our manuscript. Thanks to your crucial comments. (1) We separated the Introduction and Literature Review sections. (2) We added a subsection to clarify the hypotheses. (3) We added information on the simplified one-factor model. (4) We tempered down the claims in Study 2. (5) We improved the reproducibility of the R code. We hope to provide a better product for PLOS ONE.

Journal Requirements

Requirement 1: Ensure manuscript meets PLOS ONE's style requirements, including those for file naming.

Response: Done.

Requirement 2: Provide additional details regarding participant consent. In the ethics statement in the Methods and online submission information, please ensure that you have specified what type you obtained (for instance, written or verbal, and if verbal, how it was documented and witnessed). If your study included minors, state whether you obtained consent from parents or guardians. If the need for consent was waived by the ethics committee, please include this information.

Response: Thank you very much for reminding us about the crucial requirement. We add the information both at the very beginning of our manuscript as well in the methodological section (highlighted in yellow): “Participants were informed that the study was being conducted for scientific purposes and that their participation was both anonymous and voluntary. They were provided with details regarding the composition of the research team, along with contact information for the investigators. Participants were explicitly informed of their right to withdraw from the survey at any time. Informed consent was obtained through the first item of the questionnaire, which asked: “Do you consent to participate in this study and acknowledge that you may withdraw at any point?” Only adult participants took part in the study. No sensitive data or any information that could identify the participants was collected.”

Requirement 3: Your ethics statement should only appear in the Methods section of your manuscript. If your ethics statement is written in any section besides the Methods, please move it to the Methods section and delete it from any other section.

Response: We moved it into the Methods section.

Requirement 4: We note that this data set consists of interview transcripts. Can you please confirm that all participants gave consent for interview transcript to be published? [Additional details about consent and identifying information requirements]

Response: We did not have any interview transcripts or any open-ended questions. Our study consists of questionnaires that were rated on Likert scales.

Requirement 5: Please include captions for your Supporting Information files at the end of your manuscript, and update any in-text citations to match accordingly.

Response: We have stored our Supplementary Materials in the Open Science Framework (OSF) and refer to the doi link at the end of our manuscript. Please let us know if you have any other requirements.

Reviewer #1 Comments

Comment 1: The introduction and literature review sections currently lack clear separation and detailed elaboration. Although the research background was briefly introduced at the outset, gaps identified clearly, and key concepts such as biases and self-awareness introduced effectively, a clearer and more structured distinction between these sections would significantly improve readability and coherence.

Response: We are grateful for the Reviewer's opinion. The overall favorable opinion makes us elaborate the text as best as possible according to the remarks. We provide a separate introduction that precedes the literature review section to fulfil the call for clear separation. Introduction: “The main aim of the present research is to establish a new tool that accurately and reliably assesses human self-awareness of ego-threatening biases. The second goal directs our effort to understand the distinction between self-awareness of ego-threatening biases and self- awareness of ego-non threatening biases. Self-awareness of biases assumes metacognitive processes, meaning knowledge about one's own mental states, beliefs, and emotions. Self-awareness of ego-nonthreatening biases is defined as being aware of how self-regulatory biases and psychological rules (like classic conditioning) influence one’s own behavior. Self-awareness of ego-threatening biases (like dehumanization) analogously is defined as being aware of how the biases that negatively assert about the perpetrator (threatening the ego) influence one’s own behavior. Since previous research showed self-regulatory functions of high (vs. low) self-awareness of ego-nonthreatening biases (metacognitive self), we suspected the reverse pattern for self-awareness of ego-threatening biases. For the latter, a high level of self-awareness of ego-threatening biases may relate to potential deeper and accurate self-knowledge about the dark aspect of the self.”

Comment 2: Your literature review could benefit from deeper exploration, particularly regarding the connection between self-awareness and metacognition. Given the importance of the metacognition concept to your study, expanding the discussion on how self-awareness specifically relates to metacognition would greatly enhance the relevance and depth of your theoretical framing. Comment 3: While you have successfully introduced the broader research context, highlighted existing challenges, and summarized past literature relevant to your hypotheses, the literature review would be even stronger if you included a critical analysis of past studies. Please consider explicitly discussing the strengths, limitations, and key lessons derived from previous research. This approach would better position your study within existing literature and underscore its contributions.

Response: Thank you very much for the remarks. We have corrected our manuscript according to Reviewer's 1 suggestions. We lifted some part of text over another (highlighted in orange) and added new parts to the manuscript (in yellow) under the new heading “Literature review”:

The nature of biases

Biases arise from cognitive shortcuts, or heuristics, that are deeply ingrained in our decision-making processes and often lead to systematic errors and irrational behaviors, significantly impacting our interactions and perceptions of others [1-3]. The study of biases has long intrigued social psychologists, with pioneering work by Daniel Kahneman and Amos Tversky in the early 1970s laying the foundation for understanding the pervasive impact of heuristics on human cognition. Tversky and Kahneman [4] and Chen and colleagues [5] revealed that biases result from the fast, automatic, and intuitive thinking. Therefore, individuals are not necessarily aware of their biases.

Individuals who focus more on positive than negative information about options, for example, are biased (e.g., optimistic bias; [6]). Since biases are cognitive failures [7, 8] they influence individual behavior but also have broader implications for societal outcomes. For example, biases can affect hiring decisions, interpersonal relationships, and even policy-making [9]. When individuals become aware of their own biases, this self-awareness can affect their self-regulation [10], self-concept [11, 12], and social interactions [13]. However, different types of biases may affect individuals differently.

Thus, a separate problem is the issue of bias types itself. Research is frequently limited to one or several biases. To our best knowledge, the typology of biases is understudied [14]. Several studies focus on positive versus negative interpretation biases. Baumgardner and colleagues [15] created a measure for positive and negative interpretation biases. Hannuschke and colleagues [16] define personality-congruent perception bias as positive when it reflects a perception bias: neuroticism predicted participants’ ratings of their interaction partners’ sociability and warmth indicating a positive but not a negative bias in interpersonal perception. Positive perception bias may result from a personality-congruent contrast effect: neurotic people see themselves as less sociable than others. Authors did not find a personality-congruent selection effect (negative biases). Insufficient psychological literature on negative versus positive roles of biases in many contexts of human behavior make us enlarge the potential theoretical basis.

Mentioned above there are a few approaches that formulate distinct types of biases [15, 16]. These approaches primarily classified biases based on the mechanisms that give rise to them and their societal consequences. In our research, we propose a new typology by classifying biases based on their effects on individuals once they become aware of their biases. We identified two types of biases: ego-nonthreatening biases, conceptualized as the Metacognitive Self, and ego-threatening biases. We focus on whether awareness of the bias is positive or negative for the individual and how this awareness influences self-regulation. Positive biases can enhance social interactions and well-being as, for example, in positivity bias. Conversely, negative biases can diminish social interactions and lead to behaviors such as self-handicapping or behavioral consequences of dehumanization. Awareness of negative biases can result in self-threats and lower self-regulation, whereas awareness of positive biases fosters positive emotions, well-being, and self-regulation. We term this awareness of positive biases as the metacognitive self, based on previous literature, while our exploration of the consequences of awareness of negative biases is novel.

Thus, in our research, we propose a new typology by classifying biases based on their effects on individuals once they become aware of their biases. We identified two types of biases: ego-non threatening biases, conceptualized as the Metacognitive Self, and ego-threatening biases.

Self-awareness of ego-nonthreatening biases (metacognitive self) and ego-threatening biases

Self-awareness of one’s emotions, thoughts, and decisions imply metacognition, which is the understanding and regulation of one’s own cognitive processes. This concept dates back to Dewey [17] and has been expanded by scholars like Nelson and Narens [18]. Carden and colleagues [19] provided a comprehensive definition of self-awareness based on a systematic literature review: “Self-awareness consists of a range of components, which can be developed through focus, evaluation, and feedback, and provides individuals with an awareness of their internal state (emotions, cognitions, physiological responses), that drives their behaviors (beliefs, values, and motivations) and considers how these impact and influence others.”

Building on the foundation of self-awareness, the concept of the Metacognitive Self (MCS) has been proposed. It reflects an individual’s ability to understand and manage their cognitive processes and biases, promoting better self-knowledge and self-regulation. The Metacognitive Self is consistent with Taylor and Brown’s [20] idea that some biases can be motivationally adaptive, while also acknowledging Block and Colvin’s [21] limitations of this theory. It embraces the Aristotelian idea of the golden mean, balancing the adaptive role of psychological rules with accurate self-reflection.

Individuals with a high Metacognitive Self can better regulate their behavior by recognizing and managing their biases, leading to more consistent alignment with personal and social standards. High Metacognitive Self promotes deeper self-reflection and accurate assessment of one’s actions, which is crucial for personal growth and development. The balance achieved through Metacognitive Self helps individuals maintain motivation without falling into the traps of overconfidence or excessive self-criticism. With high Metacognitive Self, individuals can internalize self-knowledge, making it readily accessible and applicable in various situations, thus enhancing decision-making and behavior [22, 23].

Kleka and colleagues [24] expanded on the role of metacognitive self and intrinsic epistemic motivation in shaping an individual’s awareness of biases. They suggest that feedback from emotional responses and interpersonal relationships helps individuals recognize biases within their own behavior. Similarly, Brycz and colleagues [25] found that strong self-awareness of positive biases (a high Metacognitive Self) fostered participants' willingness to learn social adaptive behavior even in the face of failure in contrast to participants with a low Metacognitive Self.

Comment 4: I recommend dedicating a specific subsection to clearly outline your hypotheses. It would be particularly beneficial if you explicitly connect these hypotheses to the research gaps you aim to address, emphasizing how your research introduces novelty. Ideally, this subsection should serve as a concise summary highlighting the significance and rationale for conducting your research, clearly linking research gaps, hypotheses, and anticipated contributions.

Response: Thank you very much for the remark. We did our best to summarize and point out the rationale for providing hypotheses in a specific subsection “Problem and hypothesis”: “We have presented a theoretical background for the idea of a distinction between self-awareness of ego-threatening biases and self-awareness of ego-nonthreatening biases (metacognitive self). This distinction may help us to determine different paths and shed light on the beneficial versus harmful nature of bias awareness in creating self-construct, relations with other people, and our way of feeling and thinking. Researchers have attempted to mitigate the impact of negative biases, such as stereotyping and prejudice, on human behavior. For example, implicit bias training programs have been developed to educate individuals in influential positions about the harmful effects of implicit biases on society. Koenig-Robert and colleagues [55] demonstrated that such training can effectively reduce biases from subliminal sensory primes, with effects lasting up to one week. This highlights the importance of self-awareness of biases and targeted interventions in reducing their negative impact on both individual and societal levels. The rationale behind bias education is the belief that understanding one’s own biases can prevent future errors. However, acknowledging ego-threatening biases can lead to defense mechanisms. This is particularly true for laypersons, as opposed to trained psychologists or those who have undergone implicit bias training. Thus, we predicted positive correlations between self-awareness of ego-threatening biases with moral disengagement, dark triad, less integrated self-concept in contrast to the negative (or none) correlations between enumerate variables and self-awareness of ego-nonthreatening biases (metacognitive self).”

Thank you once more for Reviewer's 1 suggestions and support.

Reviewer #2 Comments

Comment 1: The authors' findings linking self-awareness of ego-threatening biases (SAETBQ) and moral disengagement are intriguing but potentially problematic due to methodological choices. Specifically, adopting a simplified one-factor solution despite clear statistical support for a four-factor structure risks oversimplifying complex psychological relationships, thereby influencing both the interpretation and practical implications of the results. Given the ethically sensitive nature of the conclusions drawn, the authors should explicitly acknowledge how their decision to use a single-factor model limits interpretive clarity, nuance, and generaliz

---

## [Decision Letter · Decision Letter 1]

Development and Validation of the Self-Awareness of Ego-Threatening Biases Questionnaire (SAETBQ)

PONE-D-25-01433R1

Dear Dr. Baumann,

We’re pleased to inform you that your manuscript has been judged scientifically suitable for publication and will be formally accepted for publication once it meets all outstanding technical requirements.

Kind regards,

Hidenori Komatsu

Academic Editor

PLOS ONE

Reviewers' comments:

Reviewer's Responses to Questions

**Comments to the Author**

1. If the authors have adequately addressed your comments raised in a previous round of review and you feel that this manuscript is now acceptable for publication, you may indicate that here to bypass the “Comments to the Author” section, enter your conflict of interest statement in the “Confidential to Editor” section, and submit your "Accept" recommendation.

Reviewer #1: All comments have been addressed

Reviewer #2: All comments have been addressed

2. Is the manuscript technically sound, and do the data support the conclusions?

Reviewer #1: Yes

Reviewer #2: Yes

3. Has the statistical analysis been performed appropriately and rigorously? 

Reviewer #1: Yes

Reviewer #2: Yes

4. Have the authors made all data underlying the findings in their manuscript fully available?

Reviewer #1: Yes

Reviewer #2: Yes

5. Is the manuscript presented in an intelligible fashion and written in standard English?

Reviewer #1: Yes

Reviewer #2: Yes

6. Review Comments to the Author

Reviewer #1: I am glad to see that all my comments have been addressed properly. The manuscript should meet the publication standard after proofreading. Well done.

Reviewer #2: The addition of the supplemental documentation and the open source code references (https://osf.io/vhwb5/?view_only=fa592ede448042bda9053cb1b565c5aa) anomalous datapoints provides satisfactory justification for use of the one-factor approach. The models are all statistically sound and logically developed. The justification for selection of the one-factor, once expanded upon, provides an example of how overfitting a model can functionally force statistical confidence, when in reality is diminishes the output. The inclusion of the supplemental an excellent example of how adequate documentation of code structuring, ML model development, ML model choice add the elegant transparency and nuance ML applications can bring.

Code:

bootstrap.R (https://osf.io/z5anc?view_only=fa592ede448042bda9053cb1b565c5aa) - high reproducibility and consistency with table 5,

factor_comparison (https://osf.io/95bc2?view_only=fa592ede448042bda9053cb1b565c5aa) - very high reproducibility with table 4, BN24 exclusion justified

7. PLOS authors have the option to publish the peer review history of their article (what does this mean? ). If published, this will include your full peer review and any attached files.

**Do you want your identity to be public for this peer review?** For information about this choice, including consent withdrawal, please see our Privacy Policy .

Reviewer #1: No

Reviewer #2: **Yes: ** AC Demidont

---

## [Editor Report · Acceptance letter]

PONE-D-25-01433R1

PLOS ONE

Dear Dr. Baumann,

I'm pleased to inform you that your manuscript has been deemed suitable for publication in PLOS ONE. Congratulations! Your manuscript is now being handed over to our production team.

Kind regards,

on behalf of

Dr. Hidenori Komatsu

Academic Editor

PLOS ONE